# *Halorubrum pleomorphic virus-6* Membrane Fusion Is Triggered by an S-Layer Component of Its Haloarchaeal Host

**DOI:** 10.3390/v14020254

**Published:** 2022-01-27

**Authors:** Eduardo A. Bignon, Kevin R. Chou, Elina Roine, Nicole D. Tischler

**Affiliations:** 1Laboratorio de Virología Molecular, Fundación Ciencia & Vida, Santiago 7780272, Chile; ebignon@cienciavida.org (E.A.B.); kchou@cienciavida.org (K.R.C.); 2Facultad de Medicina y Ciencia, Universidad San Sebastián, Santiago 7510157, Chile; 3Molecular and Integrative Biosciences Research Program, University of Helsinki, FIN-00014 Helsinki, Finland; elina.roine@helsinki.fi

**Keywords:** haloarchaea, S-layer, *Pleolipoviridae*, HRPV-6, membrane fusion, cell-entry, receptor

## Abstract

(1) Background: Haloarchaea comprise extremely halophilic organisms of the Archaea domain. They are single-cell organisms with distinctive membrane lipids and a protein-based cell wall or surface layer (S-layer) formed by a glycoprotein array. Pleolipoviruses, which infect haloarchaeal cells, have an envelope analogous to eukaryotic enveloped viruses. One such member, *Halorubrum pleomorphic virus 6* (HRPV-6), has been shown to enter host cells through virus-cell membrane fusion. The HRPV-6 fusion activity was attributed to its VP4-like spike protein, but the physiological trigger required to induce membrane fusion remains yet unknown. (2) Methods: We used SDS-PAGE mass spectroscopy to characterize the S-layer extract, established a proteoliposome system, and used R18-fluorescence dequenching to measure membrane fusion. (3) Results: We show that the S-layer extraction by Mg^2+^ chelating from the HRPV-6 host, *Halorubrum* sp. SS7-4, abrogates HRPV-6 membrane fusion. When we in turn reconstituted the S-layer extract from *Hrr.* sp. SS7-4 onto liposomes in the presence of Mg^2+^, HRPV-6 membrane fusion with the proteoliposomes could be readily observed. This was not the case with liposomes alone or with proteoliposomes carrying the S-layer extract from other haloarchaea, such as *Haloferax volcanii*. (4) Conclusions: The S-layer extract from the host, *Hrr.* sp. SS7-4, corresponds to the physiological fusion trigger of HRPV-6.

## 1. Introduction

Archaeal viruses are highly diverse in shape [1,2,3] and several are enveloped, including members of the *Pleolipoviridae* family [4,5,6,7]. Pleolipoviruses infect *Haloarchaea*, members of the *Euryarchaeota* phylum that live in hypersaline environments ranging from 10% salinity to salt saturation [8]. Yet, the entry mechanisms of archaeal viruses are in general poorly characterized, but it is likely that archaeal viruses, like their bacterial counterparts, recognize and attach to cell wall components, such as S-layer proteins, sugar moieties, or filamentous surface structures, such as pili [9,10,11,12,13,14]. In most haloarchaea, the sole constituent of the cell wall is the proteinaceous surface layer (S-layer) that encloses the cellular membrane and functions as a protective coat that covers the whole cell [15,16,17,18]. The S-layer is composed of multiple copies of one glycoprotein, and in some cases two, capable of self-assembling into a rigid paracrystalline surface layer with a specific lattice symmetry depending on the organism [17,19,20]. In the case of the haloarchaeal *Haloferax volcanii*, the atomic structure of the S-layer protein has been recently solved by cryoelectronmicroscopy [21]. The S-layer is organized into a hexameric array of protein subunits consisting of six highly conserved immunoglobulin-like domains; these hexamers cover the whole cell with pentameric defects [21]. Each S-layer protein is anchored to the membrane by a lipid moiety in an Mg^2+^-dependent fashion [22], which is acquired through processing of its C-terminal end by archaeosortase A (ArtA) [23,24,25]. Although the main function of the S-layer is structural, it also must allow the molecular traffic to and from the cell [16,21]. During viral infection, the S-layer and the plasma membrane represent the major barriers to overcome [26]. In the case of the Sulfolobus spindle-shaped virus-1, it was shown that S-layer depleted thermophilic *Sulfolobus solfataricus* cells become less susceptible to infection [27], suggesting a role for the S-layer in viral cell entry.

Pleolipoviruses include three genera (alphapleolipoviruses, betapleolipoviruses, and gammapleolipoviruses) of double- or single-stranded circular or linear genomes [4,5,28,29]. They produce persistent infection of their haloarchaeal hosts, and virions are released continuously without generating cell lysis [30]. The pleolipoviruses are 40–60 nm in size and have been classified by their unique architecture that lacks a nucleocapsid protein; instead, their virions include at least two major structural proteins, a membrane-associated integrated protein (VP3 or VP3-like proteins) residing inside the virion that may interact with the genome during virion assembly [31] and a spike protein (VP4, also termed as VP4-like proteins) protruding from the membrane surface, which has a role in host infection [5,30,31,32]. Unprecedentedly, experimental evidence for a membrane fusion process in the archaea domain has been shown to occur during the entry of the alphapleolipovirus, HRPV-6, into its halophilic host, *Hrr.* sp. SS7-4 [33]. This fusion process required for cell entry is reminiscent to that of eukaryotic enveloped viruses [34,35]. Interestingly, the fusion-inducing viral VP4-like spike protein, VP5, has a V-shaped fold which is different from the three well-established structural classes of eukaryotic fusion proteins [33]. To initiate the fusion process, eukaryotic viral fusion proteins must be activated by an environmental stimulus; this typically involves cellular receptor binding, acidification within endocytic vesicles, proteolytic processing, or the combinations thereof [36]. The identity of the trigger that activates the virus-cell membrane fusion by the HRPV-6 spike remains unknown, although it has been proposed that infection must require an active interplay of the virion spike with the host S-layer to accommodate its traffic towards the cellular membrane [33].

We have previously demonstrated that in vitro HRPV-6 fuses with liposomes when triggered through high temperature [33]. In this work, we analyze the S-layer extracted from the host and characterize its role on HRPV-6 membrane fusion and infection at 37 °C. We show that the S-layer extract from *Hrr.* sp. SS7-4 reconstituted onto liposomes, but not the S-layer extract of *H. volcanii,* induces HRPV-6 membrane fusion under physiological conditions.

## 2. Materials and Methods

### 2.1. Haloarchaeal Host Strains and Viruses and Growth Conditions

*Hrr.* sp. SS7-4 was previously isolated as described in [5]. We used the non-pigmented strain of *Hrr.* sp. SS7-4 [33], here termed *Hrr.* sp. SS7-4NP, that was grown aerobically at 37 °C in modified growth media (MGM) in 23% artificial salt water (SW) [37]. The *H. volcanii* H1424 strain derived from the *H. volcanii* DS2 wild-type isolate [38] was kindly provided by Thorsten Allers (University of Nottingham, Nottingham, UK). This strain was grown aerobically at 42 °C in *H. volcanii* medium, Hv-YPC, containing yeast extract, peptone, and casamino acids in 18% artificial salt water [39].

### 2.2. HRPV-6 Propagation and Purification

The HRPV-6, previously isolated by Pietilä and others [5], was produced in *Hrr.* sp. SS7-4NP grown in MGM liquid medium. Cells were removed by centrifugation (Sorvall F12, 10,000× *g* for 40 min at 4 °C) and impurities were cleared by precipitation with 6% (*w*/*v*) polyethylene glycol 6000 (PEG6000) at 4 °C for 1 h with gentle stirring followed by removal of the precipitant by centrifugation (Sorvall F12, 10,000× *g* for 40 min at 4 °C). The HRPV-6 contained in the supernatant was then concentrated by precipitation in the final concentration of 11% PEG6000 at 4 °C for 1 h with gentle stirring. Viral particles were precipitated by centrifugation (Sorvall F12, 11,000× *g* for 40 min at 4 °C) and the pellet was resuspended in 18% SW and filtered using a polycarbonate sterile syringe filter with a pore size of 0.2 μm (EDLAB). For microscopy analysis, HRPV-6 viral particles were further purified by a sucrose gradient, adding them to the bottom of the gradient and adjusting to 50% (*w*/*v*) sucrose. The additional sucrose steps of 40%, 35%, and 0% were then carefully layered on top. After centrifugation for 2 h at 300,000× *g*, the fractions were collected and the ones containing purified HRPV-6 were identified by measuring absorbance at 280 nm (Figure A1, Appendix A).

### 2.3. S-Layer Extraction

The methodology for S-layer extraction was modified from Sumper and others (1990) [40]. A total of 600 mL of stationary culture of strains *Hrr.* sp. SS7-4 or *H. volcanii* cells were centrifuged at 7000× *g* for 30 min. The cell pellets were carefully resuspended in 300 mL of 20 mM Tris-HCl pH 7.5 and 3.91 M NaCl for *Hrr.* sp. SS7-4NP or 3.01 M NaCl for *H. volcanii*. The cells were centrifuged again at 7000 × *g* for 30 min and the cell pellets were carefully resuspended in 100 mL of 110 mM EDTA, 20 mM Tris-HCl pH 7.5, and 3.91 M NaCl for *Hrr.* sp. SS7-4NP or 3.01 M NaCl for *H. volcanii*. The cell suspensions were shaken for 30 min at 37 °C, and the formed spheroplasts were collected by centrifugation for 20 min at 10,000× *g*. The supernatant containing the extracted S-layer protein was concentrated by ultrafiltration with a 30,000 nominal molecular weight cutoff for *Hrr.* sp. SS7-4NP S-layer and a 100,000 nominal molecular weight cutoff for *H. volcanii* S-layer extract (Amicon Ultra Centrifugal Filter Devices, Millipore Sigma, Merck KGaA, Darmstadt, Germany).

### 2.4. SDS-PAGE

SDS-polyacrylamide gel electrophoresis (PAGE) was performed using TGX Stain-Free Acrylamide Kit 10% (Bio-Rad, Hercules, CA, USA) [41], and the fluorescence was detected in a ChemiDoc™ MP imager (Bio-Rad).

### 2.5. LC-MS/MS

Polypeptides in the S-layer extract were analyzed at the Proteomics core unit, Viikki campus at Helsinki University as described before [33]. Briefly, the sample was desalted with PD MiniTrap G25 and eluted in 1 mL of HENN-buffer (50 mM HEPES, 50 mM NaF, 5 mM EDTA, 90 mM NaCl, pH 8.0). The desalted sample was digested with Pierce™ Glu-C protease (MS Grade, Thermo Scientific, Waltham, MA, USA) overnight followed by purification with C-18 column. The dried sample containing the digested peptides were reconstituted in 30 μL of 0.1% TFA in 1% acetonitrile. An aliquot was analyzed by Q Exactive LC-MS/MS, and the acquired MS2 scans were searched with the Sequest search algorithms in Thermo Proteome Discoverer against the theoretical proteome generated from the genome sequence of *Hrr.* sp. SS7-4NP [42] using GeneMark.hmm for prokaryotes [43].

### 2.6. Liposome Production

Total lipids were extracted from *Hrr.* sp. SS7-4NP or *H. volcanii* cells based on methanol–chloroform–water (1:2:0.8 *v*/*v*) extraction, as previously described [44]. The dry lipids were later resuspended in buffer to produce multilamellar vesicles (MLVs). Large unilamellar vesicles (LUVs) were produced by freeze thawing and extrusion through a polycarbonate filter with a pore size of 0.2 μm (Avanti Polar Lipids, Alabaster, AL, USA) [45]. 

### 2.7. R18-Labeling of HRPV-6 and Liposomes

HRPV-6 was labeled using octadecyl rhodamine B chloride (R18, O-246, Molecular Probes, Thermo Fisher Scientific, Waltham, MA, USA) at a self-quenching concentration. Approximately 1.7 × 10^13^ plaque-forming units per milliliter (PFU ml^−1^) or liposomes were mixed well with 45 µg/mL (62 µM) of R18. R18-labeled HRPV-6 virions or liposomes were separated from the excess probe by using a Sephadex G-75 column (GE Healthcare, Chicago, IL, USA). The presence of the R18 label was measured by its absorbance at 560 nm.

### 2.8. Liposome Flotation Assay

The S-layer extract was mixed with MLVs or LUVs with and without 150 mM Mg^2+^ and subsequently added to the bottom of the gradient. In the case of MLVs, the bottom fraction was adjusted to 25% (*w*/*v*) sucrose, and the additional sucrose steps of 15% and 5% were carefully layered on top. In the case of LUVs, the bottom fraction was adjusted to 50% (*w*/*v*), and the additional sucrose steps of 40%, 35%, and 0% were carefully layered on top. After centrifugation for 2 h at 300,000× *g*, the presence of the R18-labeled liposomes was measured by their absorbance at 560 nm and the presence of the S-layer protein by SDS-PAGE.

### 2.9. Transmission Electron Microscopy (TEM)

Samples were mounted onto formvar-coated copper grids (300 mesh, Ted Pella Inc., Redding, CA, USA). Samples were negative stained with 2% phosphotungstic acid (Sigma-Aldrich, Merck KGaA, Darmstadt, Germany) and analyzed in a Talos F200C TEM (Thermo Scientific) transmission electron microscope at 200 kV with support from the Unidad de Microscopía Avanzada, Pontificia Universidad Católica of Chile.

### 2.10. Virus-Cell and Virus-Liposome Fusion Assay (Lipid Mixing Assay)

Virus fusion with target membranes was monitored by fluorescence dequenching of R18-labeled virions or liposomes by standard techniques [46] and as previously reported [33]. To this end, R18-labeled HRPV-6 virions were mixed with unlabeled cells or liposomes. In the case of virus–cell fusion, *Hrr.* sp. SS7-4NP cells were used at their logarithmic growth phase (OD 0.4–0.5, adsorption at 550 nm). For virus–liposome fusion, LUVs were mixed with HRPV-6 virions in a fluorimeter cuvette under continuous stirring at the indicated temperatures. For virus–proteoliposome fusion, proteoliposomes were labeled with R18, and for their mixture, an excess of virus over proteoliposomes was established to achieve fluorescence dequenching. Therefore, a protein ratio of 4:1 (virus:proteoliposomes) was used, and the protein quantity was measured by direct absorbance at 280 nm in a spectrophotometer (NanoDrop 2000, Thermo Fisher). Fluorescence dequenching was recorded continuously every 30 s at 585 nm at an excitation wavelength of 565 nm using a fluorescence spectrophotometer (Varian Eclypse, Agilent Technologies, Santa Clara, CA, USA) with 5 and 10 nm slit width for excitation and emission, respectively. The base value at time 0 was defined as 0% lipid mixing and the maximal extent of R18 dilution was determined by the addition of Triton X-100 (final concentration 0.1%) after the lipid mixing of each condition had concluded.

### 2.11. Statistical Analysis

All statistical analyses were carried out in GraphPad Prism, version 6, and SPSS software (SPSS, Inc., Chicago, IL, USA).

## 3. Results

### 3.1. The Extraction of the Host Cell S-Layer Abrogates HRPV-6 Membrane Fusion

We have previously shown that HRPV-6 can fuse with *Hrr*. sp. SS7-4NP cells when incubated at 37 °C and with cell-free liposomes when fusion is triggered by heating to 55 °C [33]. To analyze the role of the S-layer on HRPV-6 membrane fusion, we detached the S-layer from *Hrr*. sp. SS7-4NP by EDTA treatment, generating spheroplasts [47]. These spheroplasts were then mixed with R18-labeled HRPV-6 particles, and the dequenching of the dye by lipid mixing was monitored over time at different temperatures (Figure 1A). Incubation of the virus and spheroplasts at 37 °C did not result in lipid mixing. To overcome the possible lack of the physiological fusion trigger, we increased the temperature to 55 °C. At this high temperature, HRPV-6 induced lipid mixing to over 35% with the *Hrr*. sp. SS7-4NP spheroplasts (Figure 1A). This data confirms that, as with liposomes, fusion can be triggered with spheroplasts at 55 °C and that at the optimal growth temperature of 37 °C, an intact S-layer seems to play a critical role in triggering HRPV-6 membrane fusion and infection.

### 3.2. Characterization of the Host Cell S-Layer Extract

We next analyzed the S-layer extract obtained by EDTA treatment from *Hrr*. sp. SS7-4NP by SDS PAGE and mass spectroscopy analysis. In the gel, the most prominent protein band runs with an apparent molecular weight of ~100 kDa, but other protein species below 35 kDa could also be observed (Figure 1B). The Glu-C-digested S-layer extract revealed multiple possible protein hits by mass spectroscopy (Table 1). The highest overall protein scoring of 88.7 was obtained for a gene termed, “PGF-CTERM sorting domain-containing protein” (NCBI GenBank: TKX57172), a 1083 amino acid polypeptide with a calculated molecular weight of 111 kDa. The identification of 10 unique peptides of this protein reached a peptide coverage of 12.7% (Table 1). Interestingly, when blasting this protein sequence against the SwissProt section of UniProt, several carefully curated S-layer proteins are identified (Sequence alignment in Figure A2, Appendix A). The best characterized hits include the S-layer protein from *Haloferax gibbonsii* (UniProtKB: A0A0K1IRS6) (31% protein sequence identity, 1.2 e^−92^) [49] and the paralogous S-layer proteins from *Haloarcula hispanica* (UniProtKB: G0HV85, G0HV86) (27–28% protein sequence identity, 2.5–3.2 e^−70^) [50]. Furthermore, the *Hrr.* sp. SS7-4 S-layer candidate protein includes the C-terminal tripartite-structure motif composed of the three amino acids, proline, glycine, and phenylalanine (PGF), corresponding to the archaeosortases motif characterized in *H. volcanii, Haloferax gibbonsii*, *Halobacterium salinarum*, and *Haloarcula hispanica* S-layer proteins [23,51]. Further, the PGF-motif is followed by a hydrophobic stretch that putatively serves as a transmembrane anchor prior to its cleavage and a positively charged C-terminus (Figure 1C). Other proteins of the S-layer extract revealed by the mass spectroscopy analysis all showed scores below 50 (Table 1).

### 3.3. Characterization of the Host Cell S-Layer Reconstituted onto Proteoliposomes

To develop a cell-free in vitro system, we next determined if the S-layer from this extract can be reconstituted onto artificial membranes in the presence of newly added Mg^2+^. Therefore, we prepared multilamellar lipid vesicles (MLVs) with lipids extracted from *Hrr.* sp. SS7-4NP cells and labeled them with R18 for their detection by absorbance. The R18-labeled MLVs were then mixed with the extracted S-layer from *Hrr.* sp. SS7-4NP in the presence or absence of Mg^2+^ ions. Binding of the S-layer to MLVs was next assessed by a liposome flotation assay based on those described for eukaryotic enveloped viruses [56,57]. In the absence of Mg^2+^ the S-layer extract represented by the ~100 kDa band was found at the bottom of the gradient, while the R18-labeled MLVs were detected in the upper most fractions (Figure 2A), coinciding with their individual densities [58]. However, when we incubated the S-layer-MLV mixture in the presence of Mg^2+^ and subjected them to the floatation gradient, a preponderant fraction of the ~100 kDa S-layer band localized together with the MLVs in the upper fractions (Figure 2B), confirming a Mg^2+^-dependent S-layer binding to the *Hrr.* sp. SS7-4NP lipids.

Given that MLVs include several membrane bilayers, they are suboptimal for membrane fusion experiments. Thus, we proceeded to prepare proteoliposomes based on LUVs of <200 nm in size with lipids extracted from *Hrr.* sp. SS7-4NP cells and repeated the binding of S-layer extract from *Hrr.* sp. SS7-4NP in the presence of Mg^2+^ ions and purified them through a sucrose step gradient. To establish the gradient separation, we first ran the LUVs and S-layer extract in separated gradients (Figure 2C). In the absence of the S-layer extract, R18-labeled LUVs could be identified by absorbance in the first fractions (peak at F2), while the ~100 kDa band of the S-layer extract alone was detected by SDS-PAGE principally at the bottom fraction (F11) of the gradient. When the mixture of LUVs and the S-layer extract was incubated in the presence of Mg^2+^ ions and then subjected to the gradient, the signal of R18-labeled liposomes was shifted to fractions of higher density peaking at fraction F8 and was still present in F9. The F8 and F9 fractions also showed the highest presence of the ~100 kDa S-layer protein, and only a minor amount of this protein was detected in the bottom fractions F10 and F11 (Figure 2D). Hence, the fractions F8 and F9, where the signals for liposomes and S-layer protein overlapped, are likely to correspond to proteoliposomes, the liposomes to which the S-layer extract proteins are bound. To further corroborate this notion, we inspected the different fractions of the gradient (Figure 2D) by negative staining TEM (Figure 2E). In the upper-most fraction of the lowest density (F1), non-electron-dense structures of ~200 nm were observed, corresponding to protein-free liposomes (Figure 2E). On the other hand, in fractions F8-F9, where we found the maximum S-layer and R18 signal, a high content of electron-dense structures was detected that were <200 nm in size and decorated with a two-dimensional pattern on their surface. Hence, these fractions clearly correspond to S-layer-carrying proteoliposomes. Finally, in the bottom fraction (F11), no discernible structures similar to liposomes were found (Figure 2E), congruent with the absence of the R18 signal.

### 3.4. The S-Layer Extract of Hrr. *sp.* SS7-4NP Triggers HRPV-6 Membrane Fusion

To test the role of the S-layer extract in HRPV-6 fusion, we measured lipid mixing of unlabeled HRPV-6 particles with the R18-labeled proteoliposomes in the sucrose gradient fractions, F8 and F9. We used fractions, F1, F3, and F11, as negative controls (Figure 3A). HRPV-6 lipid mixing was readily observed with the increase of R18-fluorescence induced at 37 °C when incubated with proteoliposome-containing fractions, F8 and F9, while no fusion signal could be detected when HRPV-6 was mixed with the other fractions (Figure 3A). Thus, a component of the S-layer extract or *Hrr.* sp. SS7-4NP seems to be the physiological fusion trigger for HRPV-6. To analyze the specificity of the trigger, fusion was also tested with proteoliposomes prepared with S-layer from *H. volcanii*, another haloarchaeal species not susceptible to HRPV-6 infection (Figure 3B,C). The incubation of HRPV-6 with *H. volcanii* S-layer-extract-loaded proteoliposomes did not lead to detectable lipid mixing at 37 °C, while incubation with proteoliposomes carrying the S-layer extract from *Hrr.* sp. SS7-4NP induced lipid mixing of over 20%. Also, lipid mixing with unloaded liposomes was induced at 55 °C, corroborating the system (Figure 3B). Finally, we also analyzed the role of lipids by repeating the experiments using *Hrr.* sp. SS7-4NP or *H. volcanii* S-layer with liposomes prepared from lipids extracted from *H. volcanii* cells (Figure 3C). In this case, we observed the R18 dequenching with proteoliposomes made from *Hrr.* sp. SS7-4NP S-layer and lipids from *H. volcanii*. However, this system was far less optimal, reaching lipid-mixing values which are much lower compared to liposomes prepared from host lipids.

Finally, we inspected the membrane fusion results by negative stain TEM (Figure 3D). When we analyzed HRPV-6 alone, spherical particles of ~40–60 nm in diameter and irregular protrusions could be observed representing typical viral structures. When HRPV-6 was incubated at 37 °C for 30 min with proteoliposomes including the *Hrr.* sp. SS7-4NP S-layer extract (fraction F9, Figure 2D), huge structures were observed in which electron-dense spherical particles seem to be connected with diffuse, less electron-dense structures reminiscent of multiple possible fusion events with proteoliposomes. When, as a control, we incubated HRPV-6 with *Hrr.* sp. SS7-4NP S-layer extract only, electron-dense intact viral particles contained mainly in aggregates were visualized (Figure 3D).

As a whole, the data from the quantitative fluorescence analysis and from qualitative TEM inspection provide evidence that at 37 °C, HRPV-6 fusion with membranes is specifically triggered through the *Hrr.* sp. SS7-4NP S-layer extract.

## 4. Discussion

Viruses have developed various strategies to cross the host-cell envelope and infect their target cell. These vary depending on the structural and biochemical nature of their host [9,59,60]. Although there is relatively little information about the entry of archaeal viruses into host cells, it seems that some archaeal viruses rely on processes analogous to bacterial or eukaryotic virus counterparts [10,33]. However, since the haloarchaeal host-cell structures serving as receptors for viruses have not been fully characterized, a more complete comprehension of virus–host interactions in archaea is paramount to understand the evolutionary pressures on viruses to adapt to their cellular receptors. The haloarchaeal S-layer is commonly mentioned as one of the most probable viral receptors, considering its role as the first barrier that viruses must overcome before they can penetrate the cellular membrane [9,10,26]. Based on the molecular structures obtained for haloarchaeal pleomorphic viruses, a particular mechanism has been proposed; the V-shaped VP4-like protein is therein believed to change its conformation into an elongated structure upon binding to the host S-layer [33]. Such an extended form may allow the spike protein to reach the membrane across the ~10 nm thick S-layer [18,21]. In this work we provide evidence that a component of the S-layer extract induces HRPV-6 membrane fusion, thereby triggering the activation of the viral VP4-like fusion protein.

The principal component in our S-layer extract was a ~100 kDa protein that corresponds most likely to the S-layer protein of *Hrr.* sp. SS7-4. It was identified as PGF-CTERM sorting domain-containing protein by mass spectroscopy. It contains a typical N-terminal signal sequence for export through the cytoplasmic membrane and characteristic PGF-motif for processing by archaeosortase A followed by a predicted C-terminal transmembrane domain and residues of positive charge. In the presence of Mg^2+^ the ~100 kDa protein bound to liposomes and the TEM analysis show its arrangement into a two-dimensional array; all typical characteristics of S-layer proteins. While BLAST similarity searches against GenBank give high identity matches to several hypothetical proteins or PGF-CTERM sorting domain-containing proteins, more thorough sequence analysis was conducted with experimentally verified S-layer protein sequences of haloarchaeal species *Haloferax gibbonsii* and *Haloarcula hispanica* [49,50]. Together, the sum of sequence properties and experimental evidence support the conclusion that the ~100 kDa protein is an S-layer protein of *Hrr.* sp. SS7-4NP. However, although the S-layer extract was highly enriched in the ~100 kDa S-layer candidate protein, the ultimate identity of the protein triggering HRPV-6 fusion remains to be confirmed. Additionally, it cannot be ruled out that a secondary trigger present in the S-layer extract may complement the activation mechanism.

Other proteins that were identified in the S-layer extract correspond to a thermosome subunit protein that was recently suggested to be membrane associated [52] and whose active form is dependent on Mg^2+^ ions and ATP [53]. Also, we identified a class II fumarate hydratase that is part of the electron transport chain and in bacteria has been found to be located in the periplasm in both soluble form and bound to the membrane [55]. The other found protein corresponds to the ABC transporter substrate-binding protein which is a periplasmic, membrane-bound protein [54] that is lipid modified during its export through the cellular membrane to anchor it [61,62]. In this sense, the identified protein sequences correspond to proteins that are associated with the cellular membrane and could easily be drawn into the S-layer extract.

It is interesting to notice, that apparently a high density of S-layer extract incorporated onto the proteoliposomes is required to induce fusion efficiently. Proteoliposomes in the lower fraction (F9) of the floatation gradient induced over 80% of fusion, while fractions of lower density (F8) reached 40% of fusion with the same viral preparation and within the same time frame. Fusion of HRPV-6 was not detectable with fractions of still lower density, although they contained liposomes and some S-layer. Such a phenomena may be related to the multimerization state of the S-layer into a multimeric array [15,16,17,21,63], which may be required to efficiently trigger the fusion protein of HRPV-6.

To our knowledge, this is the first time that a virus receptor system of the archaea could be reconstituted in vitro. With this system we have reconstituted the S-layer system, common to many archaea, allowing to clearly identify the minimal components triggering HRPV-6 membrane fusion. This in vitro system has the potential to greatly increase the characterization of additional viruses and the interactions of S-layer proteins in archaea in general.

## Figures and Tables

**Figure 1 viruses-14-00254-f001:**
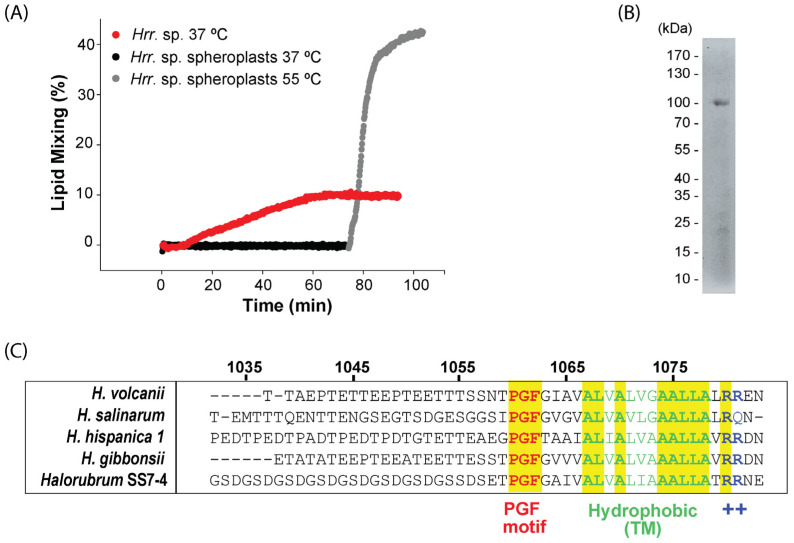
Fusion of HRPV-6 with *Hrr.* sp. SS7-4NP cells or spheroplasts and characterization of the *Hrr.* sp. SS7-4NP S-layer. (**A**) HPRV-6 fusion kinetics with *Hrr.* sp. SS7-4NP cells and spheroplasts. R18-labeled HRPV-6 was incubated with *Hrr.* sp. SS7-4NP host cells at 37 °C (red dots) or with *Hrr.* sp. SS7-4NP spheroplasts at 37 °C (black dots) or at 55 °C (grey dots) under continuous stirring. Fluorescence dequenching induced by lipid mixing was measured at 585 nm using an excitation wavelength of 565 nm. The experiments are representative for *n* = 2 biological replicates. (**B**) The protein profile of the extracted S-layer from *Hrr.* sp. SS7-4NP. SDS-PAGE of the S-layer extract visualized through fluorescence detection. (**C**) Multiple sequence alignment was run by Jalview [48] and view of the alignment at the C-terminal region of the *Hrr.* sp. SS7-4 S-layer candidate protein (PGF-CTERM sorting domain-containing protein; NCBI GenBank: TKX57172) compared to the S-layer proteins from *H. volcanii, Halobacterium salinarum, Haloferax gibbonsii*, and *Haloarcula hispánica* (UniProtKB Entry numbers: P25062, A0A4D6GUB7, ABY42_04395, and G0HV86). The five sequences show the PGF motif followed by a hydrophobic region predicted to be a transmembrane region and a positively charged region typical of the substrates of archaeosortases [23,24,25]. Amino acid positions refer to the sequence from *Hrr.* sp. SS7-4.

**Figure 2 viruses-14-00254-f002:**
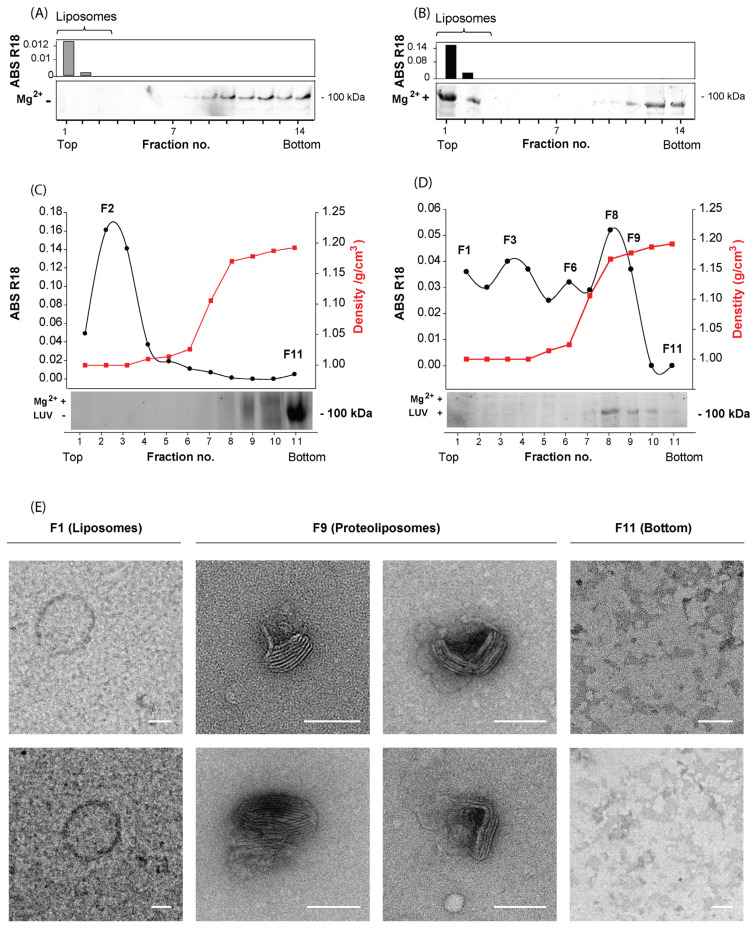
Detection and morphological characterization of S-layer-extract-loaded proteoliposomes. (**A**,**B**) Sucrose-density flotation of R18-labeled MLVs with S-layer extract in absence (**A**) or presence (**B**) of Mg^2+^ ions followed by gradient separation. Analysis of fractions from the sucrose gradient by SDS-PAGE and R18 absorbance. (**C**,**D**) Sucrose-density floatation of the LUVs and S-layer extract separately (**C**) or after mixing in presence of Mg^2+^ (**D**) followed by gradient separation. Analysis of fractions from the sucrose gradient by SDS-PAGE, density (red squares), and R18 absorbance (black circles). (**E**) Negative-stain TEM analysis of fractions F1, F9, F11 obtained from the experiment indicated in panel (**D**). The white bar indicates 100 nm. All experiments shown are representative for *n* = 2 biological replicates.

**Figure 3 viruses-14-00254-f003:**
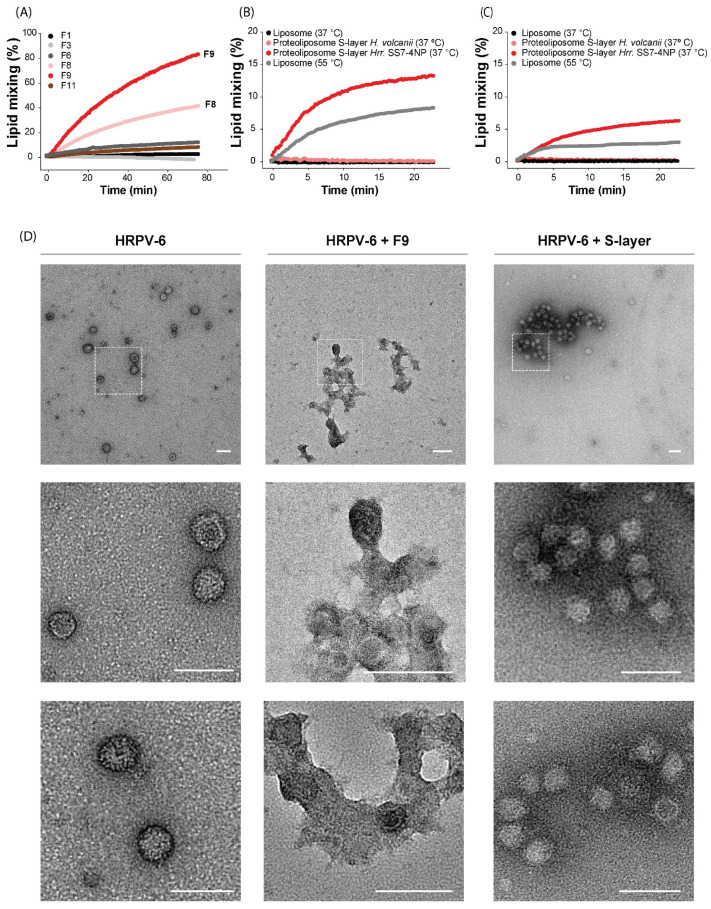
Fusion of HRPV-6 with proteoliposomes carrying the *Hrr.* sp. SS7-4NP S-layer. (**A**–**C**) Lipid mixing of HPRV-6 was measured at 37 °C or 55 °C with the indicated targets by excitation at 565 nm and measuring R18 dequenching at 585 nm. (**A**) Quantification of HRPV-6 lipid mixing with different fractions from the gradient (Figure 2D) corresponding to liposomes loaded with S-layer extract from *Hrr*. sp. SS7-4NP. (**B**) Quantification of HRPV-6 lipid mixing with liposomes loaded with S-layer extract from *Hrr.* sp. SS7-4NP or *H. volcanii*. (**C**) Quantification of HRPV-6 lipid mixing with liposomes prepared with lipids from *H. volcanii* and loaded with the S-layer extract from *Hrr.* sp. SS7-4NP or *H. volcanii*. The experiments are representative for n = 2 biological replicates. (**D**) Negative-stain TEM analysis of HRPV-6 alone or incubated either with proteoliposomes carrying the *Hrr.* sp. SS7-4NP S-layer extract (purified fraction F9, Figure 2D) or with the *Hrr.* sp. SS7-4NP S-layer extract alone. In the first row, a representative panoramic view of each condition is shown, and the square indicated with dotted lines is magnified in the second row. The third row shows representative magnifications from different images. The white bar indicates 100 nm.

**Table 1 viruses-14-00254-t001:** Main proteins identified in the S-layer extract by mass spectrometry.

Protein Name	Accession/Locus Tag Numbers (NCBI/UniProtKB)	Length (aa)	MW [KDa]	Score	Sequence Coverage (%)	Unique Peptides	Cell Localization
PGF-CTERM sorting domain containing protein	TKX57172/ EXE44_11260	1083	111.4	88.7	12.7	10	Periplasm (this paper)
Thermosome subunit protein	TKX57972/ EXE44_08630	550	57.7	47.9	23.4	10	Membrane/ cytoplasm [52,53]
Carbohydrate ABC transporter substrate-binding protein	TKX56931/ EXE44_12795	432	46.0	43.5	15.7	4	Periplasm [54]
Class II fumarate hydratase	TKX59009/ EXE44_05545	469	49.9	39.8	24.1	8	Periplasm [55]

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
