# Peer review of "Halorubrum pleomorphic virus-6 Membrane Fusion Is Triggered by an S-Layer Component of Its Haloarchaeal Host"

_viruses, 2022, doi:10.3390/v14020254_

Round 1

Reviewer 1 Report

The authors convincingly show that the receptor for Halorubrum
pleomorphic virus-6 (HRPV-6) is present in a S-layer extract
from the host Halorubrum sp. SS7-4. This nicely expands on a
previous study (also co-authored by ER and NDT) that reports
lipid fusion as the entry mechanism of HRPV-6. In the current
study, the series of well-designed experiments culminates in the
development of an in-vitro assay for interaction of a virus with
a component from its host.

I did not detect any serious shortcomings in this study, but see
the potential to improve the manuscript in various ways. The
points which I raise, none of which is severe, are sorted
(and enumerated) by position in the manuscript, not by relevance.

[1] introduction: the authors use the virus HRPV-6 and as
host strain, either the natural host (Halorubrum sp. SS7-4) or
a mutagenized variant However, I did not detect where in the
introduction the isolation of virus and host is reported
(it is Ref 6, which in the introduction is cited, but only
in a broad context). In case the authors used the mutagenized
version, this is reported in Ref 29 (PMID:30783086).
- line 81 "in the low-pigment Hrr. sp. SS7-4". This statement
makes me question that the original isolate was used. It reads
as if the "non-pigmented mutant strain" reported in Ref 29 was
used. In that case: it seems not appropriate to use the same
name (SS7-4) for an original isolate and for a mutant where
EMS mutagenization. This mutagenization probably leads to
numerous genome alterations!

[2] line 35: references are [15-18]. I would recommend to add
PMID:34818541 (complete atomic structure of a native
archaeal cell surface) as an additional reference
- this paper could also be cited for a statement in line 42,
where currently only Ref 16 is cited
- the authors should check if this paper reports the thickness
of the S-layer (see line 318)

[3] line 41: for the statement made in that sentence, namely that the
S-layer glycoprotein is C-terminally processed by artA, the authors
cite references 22 and 23. Ref 23 (PMID:28069824) is, however, mainly
concerned with artA processing of a different protein (HVO_0405)
while e.g. PMID:23651326 and PMID:26712937 (from the same group) are
primarily concerned with C-terminal processing of the
S-layer glycoprotein.

[4] line 74: the only header in the manuscript which terminates
with a dot

[5] line 76: "H. volcanii H1424 strain". It could be made clearer
that this laboratory strain is derived from the DS2 wildtype and
not an independent isolate. Strain DS2 is a wildtype isolate and
the type strain of Haloferax volcanii. Several intermediate steps,
one plasmid curation and several one-gene deletions, result in
strain H1424.
- It can be questioned if this should be referred to as
"H. volcanii H1424" throughout the manuscript, or simply as
"H. volcanii", given that the actual strain (H1424) is clearly
described in the Methods and it is stated that this strain has
been used throughout.

[6] line 83: "w v-1": either w/v or the -1 must be superscript
(similar in lines 123, 130)
(in lines 137, 139 w/v is used)

[7] line 84: "followed by removal of the precipitant by
centrifugation". As the virus is in the precipitant, a term
like "collection" may be preferrable over "removal". Or does
the virus not precipitate at 6% but only at 11%? in which case
"was concentrated by precipitation with" would be misleading.

[8] line 93: "and purified HRPV- identified": make it HRPV-6
- is "identified by SDS-page" documented in any way in
the manuscript?

[9] line 108: "TGX staining" seems self-contradictory because the
system is announced by Bio-Rad as 'stain-free'.
- same in line 204, 255

[10] line 112: "was analyzed at the Proteomics Core Unit,
Viikki campus at Helsinki University as described before [13]".
The title of that paper starts with "Haloarchaeal myovirus phiCh1",
is from the Witte lab, and does not report any proteomics experiment.

[11] line 118: should ACN be abbreviated or spelled out as
acetonitrile?

[12] line 119: "against home-made protein database of
Hrr. sp. SS7-4 [38]" What is 'home-made' here? The genome of
'Halorubrum sp. SS7' has been sequenced in the context of Ref_38
(as is evident from GenBank accession SGXV01000011.1, which is
one of the contigs from a WGS genome project summarized
under accession SGXV01000000). However, genome sequencing was
done by another group and thus is not 'home-made'. The extraction
of the theoretical proteome from this genome might be 'home-made'.
Please clarify.

[13] line 136: "the bottom fraction was and adjusted to" should be
"was adjusted to"

[14] line 140: "the presence of R18-labelled liposomes
was measured was measured by its absorbance at"; duplication
and liposomes (plural) requires "their absorbance"

[15] line 146: "transmission microscope" might be
"transmission electron microscope"

[16] line 151: "To this end, HRPV-6 virions were mixed with
cells or liposomes." In line 127 it is stated that HRPV-6 and
liposomes were labelled. If I understand correctly, the
experiment referred to in line 151 makes only sense if
labelled HRPV-6 was mixed with unlabelled liposomes. This
should be stated more clearly. The labelled liposomes are
used in another experiment (described in lines 220ff).

[17] line 160-177: The information flow seems not optimal.
However, I want to make clear that I consider the experiment
itself, the data, and the interpretation of the data as
being unambigous.
(a) I think that the results "as previously described" should
be mentioned first (intact cells), followed by the novel experiment
(spheroplasts)
(b) the sentence in lines 171-173 should be reworded, e.g.
"As we have shown before with a cell-free liposome system [29],
a temperature increase to 55 °C allows HRPV-6 induced lipid mixing,
possibly by overcoming the lack of the physiological fusion trigger"
(c) then it could be stated that this effect is also seen with
spheroplasts
(d) was the 55°C experiment also performed for intact cells?
If yes: what were the results; if not: why?

[18] line 183: 88,7 should be corrected to 88.7

[19] line 185: please do not only provide the NCBI protein
accession (TKX57172) but also the locus tag (EXE44_11260)

[20] lines 187-196: the authors argue by similarity to the
S-layer glycoprotein from H.volcanii and Halobacterium salinarum
that this is most likely the S-layer glycoprotein of the host strain.
Upon BLAST with UniProt:A0A4V6E361 (which is retrieved when searching
for TKX57172 or EXE44_11260) against the SwissProt section of UniProt,
several characterized S-layer glycoproteins are identified.
(a) the best hit is from Haloferax gibbonsii (UniProt:A0A0K1IRS6)
(31% protein sequence identity, e-93; the alignment is in two blocks,
probably due to some internal duplication). Characterization of the
H.gibbonsii protein is described in PMID:29558455.
(b) The next BLASTp hits (28% protein sequence identity, e-71) are
two paralogous S-layer glycoproteins from Haloarcula hispanica,
with characterization being described in PMID:26170448.
(c) Together, these homologs support the conclusion of the authors
much more strongly than the currently described similarity, which
is restricted to the ArtA targetting region
- this also applies to the discussion, line 322ff

[21] line 189: species names require italics

[22] line 201: "Fluorescence dequenching induced by lipid mixing was
measured at 585 nm". For a fluorescence experiment, the excitation
wavelength should also be specified.
- also applies to Figure 3 legend

[23] line 223: "an important fraction". It seems that 'important' is
meant semi-quantitatively, but that seems an invalid usage of the
word 'important'.

[24] lines 254,256: "and gradient separation" might be "followed by
gradient separation"

[25] line 213ff: It may be preferrable to mention two experimental
details prominently right at the beginning of this paragraph, rather
than providing this information just in a half-sentence "by the way":
(a) MLVs are labeled with R18 (currently mentioned only in line 220
"while the R18-labeled MLVs were detected", not mentioned in
line 215 "we prepared multilamellar lipid vesicles (MLVs) with
lipids extracted from ...")
(b) in this experiment (in contrast to others in the same
manuscript) only the absorption of R18 is exploited, not its
fluorescence  (currently pointed to only in line 220 "were
detected by absorbance").

[26] line 236: "peaking at fractions F8 to F9". To also name
fraction F9 is rather 'generous' ('ABS R18' of F9 corresponds to that
of F3 and F4). Clearly, the peak of 'ABS R18' and of the 100 kDa protein
is in F8. However, the authors subjected F9, not F8, to TEM analysis.
(F8 was used for other, and more relevant, experiments). But, as the
authors correctly point out, also F9 shows correlation of 'ABS R18' and
a strong band at 100 kDa, and thus the experiment is convincing in
its current form.

[27] Fig.2(line 251): "F9(Proteoiposomes)" lacks the "l" of liposomes

[28] line 262 ff: "we measured lipid mixing of HRPV-6 particles with
the R18-labelled proteoliposomes". I am surprised that the
experiment works in this direction. When virus is labelled at
self-quenching concentration, then membrane fusion of the rather
small virus with the rather large cellular membrane results in a
sufficiently large dilution so that self-quenching is overcome.
However, when the liposome is labelled, the amount of lipid added
due to the fused virus may be quite low, so that de-quenching might
not occur or might lead to only a very weak signal. Please comment.

[29] line 262: "we measured ... fractions F8 and F9 (Figure 2D)". That
reads as if Fig.2D would show the result of the mixing experiment,
which is not the case, so that the statement is misleading. Fig.2D
shows what 'fractions F8 and F9' refers to. However, as this is
described just a few lines above, I think that "(Figure 2D)" can
be left out.

[30] line 264: "We used fractions F1, F3 and F11 as negative
controls (Figure 4A)." Please correct to Figure 3A
- same error in line 271 "Figure 4B-C", line 275 "Figure 4B"

[31] line 276: "by repeated the experiments" should be "by repeating"

[32] line 283ff(Figure 3 legend): this legend should be improved
(a) the authors add the panel info (A) (B) (C) only after stating
what is shown. There is (A-D) before that, but it is enigmatic
what this refers to. And anyhow, if it refers to "HPRV-6 fusion
kinetics", is needs to be A-C, not A-D. It may be better to
start for each panel with its label and then describe what
is displayed.
(b) for (B), it is stated "loaded with S-layer extract from
H. volcanii H1424" but the data for the S-layer extract of
the native host are also shown, as are the 55 °C data
(c) "was mixed ... at 37 °C or 55 °C" might be "was mixed ... and
incubated at..."

[33] line 299: "with sharp boarders": correct to "borders". But what
is meant, that the individual virus particles have a sharp border,
or that the aggregates do not 'fade out' into the vicinity but seem
to be rather well 'encapsulated'?
- I like the figure provided as "19" in the extra images; this
might be shown as a Suppl.Fig. because it very nicely illustrates
how many virus particles densely concentrate on 'something',
while there are just a few additional virus particles located
in the vicinity.

[34] line 331: "were identified in the S-layer extracted correspond to"
should be "S-layer extract"

[35] line 338: the authors might not only cite Ref_51 but
also PMID:20886060

[36] line 343: "the lowest fraction (F9) of the floatation gradient"
should be "the lower fraction" (because the lowest fraction, F11,
does not contain proteoliposomes.

[37] line 355ff: remove dots after NDT and KRC
- correct EB to either ER or EAB.

Reviewer 2 Report

General comments

In this manuscript, Bignon and coworkers show that HRPV-6 virus requires its host S-layer protein to fuse with the host membrane. The system that they develop with S-layer containing proteoliposomes is innovative and potentially applicable to other archaeal virus-host systems, which is a big plus. Since virtually all archaeal S-layer proteins are N-glycosylated it would have been nice to enzymatically de-glycosylate the S-layer and test the effect on fusion, but this may be difficult to do, especially given that these N-glycosylations have probably not been characterized in the Halorubrum host.

An easier experiment to do will, although less interesting, will be to heat shock H. volcanii spheroplasts and show whether the virus then fuses with them. However, this is not a must.

Specific comments

Clearly temperature increase is very significant for envelope biology of the Halorubrum host strain, but one wonders whether this will also be applicable to other haloarchaea. Where was this species isolated from and what is its growth temperature optimum and maximum?

Line 12 "Its member, Halorubrum pleomorphic virus-6 (HRPV-6)" – rephrase to “Line 12 "One such pleolipovirus, Halorubrum pleomorphic virus-6 (HRPV-6)"

Line 22

“extract from other haloarchaea” – should be “extract from the species Haloferax volcanii

Line 38

“haloarchaeal Haloferax volcanii,” – should be “haloarchaeal model organism Haloferax volcanii,”

Line 76

“H. volcanii H1424 strain” – if this strain was chosen for a particular reason, such as lack of pigments to facilitate fluorescent microscopy this should be stated here.

Line 129

“Approximately 1.7 × 1013 plaque forming units per milliliter (PFU ml-1)” – fix numbers, also does this virus form plaques or was this measured by particle count?

Line 189 – species names should be in italics

Line 331 – a recent paper (https://doi.org/10.1371/journal.pbio.3001277) claims that in Haloferax thermosome subunits can also be membrane associated, which is very relevant here.
